# Toward the Eradication of Herpes Simplex Virus: Vaccination and Beyond

**DOI:** 10.3390/v16091476

**Published:** 2024-09-17

**Authors:** Jane Y. Chang, Curt Balch, Hyung Suk Oh

**Affiliations:** 1Ascendant Biotech Inc., Foster City, CA 94404, USA; 2Bioscience Advising, Cincinnati, OH 45208, USA; curt.balch@gmail.com; 3Department of Microbiology, Blavatnik Institute, Harvard Medical School, Boston, MA 02115, USA; hyungsuk_oh@hms.harvard.edu

**Keywords:** herpes simplex virus type 1, live-attenuated vaccines, mRNA vaccine, replication-defective strains, interferon response

## Abstract

Herpes simplex virus (HSV) has coevolved with *Homo sapiens* for over 100,000 years, maintaining a tenacious presence by establishing lifelong, incurable infections in over half the human population. As of 2024, an effective prophylactic or therapeutic vaccine for HSV remains elusive. In this review, we independently screened PubMed, EMBASE, Medline, and Google Scholar for clinically relevant articles on HSV vaccines. We identified 12 vaccines from our literature review and found promising candidates across various classes, including subunit vaccines, live vaccines, DNA vaccines, and mRNA vaccines. Notably, several vaccines—SL-V20, HF10, VC2, and mRNA-1608—have shown promising preclinical results, suggesting that an effective HSV vaccine may be within reach. Additionally, several other vaccines such as GEN-003 (a subunit vaccine from Genocea), HerpV (a subunit vaccine from Agenus), 0ΔNLS/RVx201 (a live-attenuated replication-competent vaccine from Rational Vaccines), HSV 529 (a replication-defective vaccine from Sanofi Pasteur), and COR-1 (a DNA-based vaccine from Anteris Technologies) have demonstrated potential in clinical trials. However, GEN-003 and HerpV have not advanced further despite promising results. Continued progress with these candidates brings us closer to a significant breakthrough in preventing and treating HSV infections.

## 1. Introduction

Herpes simplex virus (HSV) ranks among the oldest and most successful vertebrate pathogens, noted for its ubiquity, lifelong persistence in hosts, and evolutionary codivergence spanning hundreds of millions of years [1]. The World Health Organization (WHO) estimates that 3.7 and 0.5 billion people worldwide are infected with HSV-1 and HSV-2, respectively [2,3]. In the United States alone, annual expenditure on HSV treatment reaches USD 8 billion, not counting productivity losses [4]. HSV, a large, enveloped DNA virus, infects epithelial surfaces lytically and establishes lifelong latency in sensory neurons. Despite often remaining asymptomatic during latency, HSV can cause recurrent outbreaks manifesting as epithelial blisters and cold sores, with severe complications including encephalitis, meningitis, and vision-threatening herpetic keratitis, the leading cause of infectious blindness [2,3].

Efforts to find a permanent solution to HSV have been a focus of scientific and medical research for over a century, with the first published vaccine attempt dating back to 1926 [5]. However, as of 2024, no effective functional cure, vaccine, or method to eliminate lifelong infection has been developed. The widespread prevalence of HSV poses significant challenges and offers insights into superinfections, such as those involving HIV. It is estimated that a successful HSV vaccine could reduce HIV incidence by 30–40% over 20 years [6]. Additionally, due to HSV’s infection of peripheral and CNS neurons, it has been linked to neurodegenerative diseases like Alzheimer’s disease [7,8].

Addressing the economic and medical challenges posed by HSV, including its increasing incidence, underscores the urgent need for the development of both prophylactic and therapeutic vaccines. Recent innovative approaches, such as nanotechnology and genomic editing, also show promise. In 2016, the WHO’s Product Development for Vaccines Advisory Committee emphasized the critical need for HSV vaccines for global use. Concurrently, the Global Health Sector Strategy for Sexually Transmitted Infections identified vaccine development as a key innovation for the future control of sexually transmitted infections [9].

## 2. Methods

Two authors (J.Y.C. and C.B.) conducted a comprehensive literature search using PubMed, EMBASE, Medline, and Google Scholar with the search terms “vaccines and HSV”, “vaccines and herpes simplex virus”, “herpes simplex virus immunization”, and “HSV immunization”. They independently screened the abstracts for clinically relevant articles. Any disagreements regarding the inclusion of articles for specific therapeutic and preventive approaches were discussed and resolved prior to their incorporation into the review, ensuring relevance and historical support.

### 2.1. HSV Life Cycle, Pathogenesis, and Immune Evasion

HSV-1 and HSV-2, primarily responsible for infections of the orofacial and genital mucosal surfaces, respectively, share a common structure. Each virus consists of a large, double-stranded, linear DNA genome (encoding over 84 genes) housed within an icosahedral protein cage, or nucleocapsid, which is enveloped by a lipid bilayer [10]. The nucleocapsid is connected to the envelope by “teguments”, unique structural proteins encoded by the virus. The entire particle is referred to as a “virion” (Figure 1). HSV genes encode various proteins essential for forming the capsid, teguments, and viral envelope, as well as for controlling the replication and infectivity of the virus. The genomes of HSV-1 and HSV-2 are complex and divided into two segments: unique L (UL) and unique S (US). These segments encode 57 UL and 12 US single-copy genes separated by repeated genes [10,11]. The UL and unique US regions are each flanked by internal and terminal inverted repeats (Figure 1). Several key genes are located in repeat regions and are diploid.

The transcription of HSV genes is catalyzed by RNA polymerase II of the infected host [12]. HSV follows a cascade gene expression pattern, progressing from immediate early (IE) to early (E) and late (L) periods. IE genes, which are expressed first, encode proteins that regulate the expression of E and L viral genes. E genes encode enzymes involved in DNA replication and the biosynthesis of envelope glycoproteins. The expression of L genes, which occurs last, predominantly encodes proteins that form the virion particle [10], ultimately leading to the egress of mature virions from the host cell.

The entry of HSV into cells requires the coordinated interaction of a variety of glycoproteins found on the virion surface [13]. The initial attachment of a virus to a host cell is mediated by multiple viral glycoproteins and various binding receptors [13]. Although the fusion machinery is similar for all herpesviruses, each species uses distinct receptors and receptor-binding glycoproteins. Key glycoproteins targeted by HSV vaccines include glycoprotein C (gC), glycoprotein D (gD), and glycoprotein E (gE). gD, which is essential for virion infectivity, is the required receptor-binding protein for most alphaherpesviruses. gD binds to three classes of receptors, including nectins, herpesvirus entry mediators, and a modified form of heparan sulfate [13]. Nectin-2 mediates the entry of some HSV-1 and HSV-2 strains, while derivatives of heparan sulfate can also serve as entry receptors for HSV-1 [13]. Meanwhile, gC, which is required for the efficient binding of virions to the cell surface, is implicated in the infection process [14]. Finally, gE binds the Fc domain of immunoglobulin G, contributing to HSV immune evasion [15].

Initial expression of the five IE genes, ICP0, ICP4, ICP22, ICP 27, and ICP 47, is driven by the viral transcription factor VP16, a virion tegument protein [16]. HSV-1 encodes its own RING-finger E3 ubiquitin ligase in the form of infected cell polypeptide 0 (ICP0), which directly interfaces with component proteins of the ubiquitin pathway to inactivate host immune defenses [17]. Consequently, ICP0 plays a critical role in the infectious cycle of HSV-1, which is required to promote the onset of lytic infection and the reactivation of viral genomes from latency [17]. Furthermore, mutations that inactivate ICP0 render HSV-1 highly sensitive to interferon inhibition [18].

The IE gene product ICP4 (encoded by *RS1*) is one of the major regulatory factors required to efficiently activate the transcription of early and late viral genes during HSV infection [19]. ICP4 is a transcriptional regulator that plays a prominent role within this cascade [20]. It carries out these functions by interacting with DNA and modulating host cellular RNA polymerase II activity on viral genes [21]. ICP27 (encoded by *UL54*) is another essential IE protein that upregulates early and late viral genes [16]. ICP27 performs multiple functions during HSV-1 infection, many of which involve post-transcriptional effects on viral and cellular mRNAs [16]. ICP4 and ICP27 interaction is required to properly incorporate ICP4 into the virion. ICP22 (encoded by *US1*), necessary for efficient acute replication in mice, appears to be the least studied of the five IE proteins [22].

Other important HSV proteins include ICP8 (encoded by *UL29*), a replicative single-stranded DNA binding protein that forms filaments and binds cooperatively to single-stranded DNA. ICP8 interacts with other viral and cellular proteins to promote viral DNA replication [23]. Tegument proteins are a group of viral structural components that play an important role in viral gene replication and virion assembly [24]. Several HSV-1 tegument proteins have been identified as being associated with viral DNA transport into and out of the nucleus for viral assembly [24]. The latency-associated transcript (LAT) is an unusual region of the HSV genome, located in the long repeat region (TR_L_ and IR_L_) (Figure 1B), which is not silenced by viral genes during latency. Genetic studies involving deletions and mutations within the LAT promoter and 5′ exon-coding region have yielded a multitude of phenotypes associated with anti-apoptosis, neuronal tropism, reactivation, leakiness of lytic gene transcription during latency, and histone H3 methylation [25]. Finally, the mechanism of action of acyclovir requires phosphorylation by the viral thymidine kinase (encoded by *UL23*) to yield acyclovir monophosphate, which then eventually induces the repression of viral replication, through reduced DNA polymerase activity [26].

Productive infection results in the formation of vesicular lesions in the mucosal epithelia. This is followed by viral spread to the axonal terminals of peripheral neurons, retrograde transport toward the cell body, and the establishment of a latent infection that typically persists for the life of the host. Reactivation leads to the anterograde transport of viral progeny and the reinfection of the innervated epithelial tissues [27]. Reactivation can be triggered by cellular (nutritional or oxidative stress) or environmental (emotional stress, fever, UV exposure, hormonal changes, surgery, cranial trauma, etc.) factors. However, it remains unclear whether these triggers act directly on infected neurons or through systemic functions [27].

### 2.2. HSV Defense Mechanisms to Evade Immune Responses

HSV has developed numerous defense mechanisms to evade both innate and adaptive immunity. As depicted in Figure 2, the HSV UL41 endoribonuclease vhs degrades mRNA encoding the cellular component cGAS. Simultaneously, the HSV ICP27 protein inhibits the serine–threonine kinase TBK1, which is crucial for the cell’s inflammatory response by phosphorylating the oligomerized stimulator of interferon genes (STING). This inhibition prevents the recruitment of TRAF6, thereby blocking the activation of nuclear factor-kappa B (NF-κB) and suppressing the inflammatory response (Figure 2). Additionally, HSV ICP27 inhibits interferon regulatory factor 3 (IRF3), which can also activate NF-κB [28]. The HSV proteins UL42 and HSV-1 γ34.5 further inhibit the nuclear translocation of the p50 and p65 subunits of NF-κB, thus suppressing the activation of inflammatory genes [29].

Herpesviruses also hijack component proteins of the host ubiquitin machinery to subvert cellular processes and promote replication [17,30]. During HSV-1 infection, these events are largely driven by the E3 ubiquitin ligase ICP0, which promotes the successful onset of lytic infection and the productive reactivation of viral genomes from latency (see Section 2.1). It has been suggested that identifying and developing inhibitors to ICP0 could be beneficial in treating recurrent HSV-1 infections by allowing host immune defenses to block viral reactivation from latency [29].

### 2.3. Current HSV Therapy

The synthetic acyclic guanosine analog acyclovir (ACV), discovered in the 1970s [31], remains the first-line agent for the prophylaxis and treatment of HSV infections. ACV is phosphorylated by the virally encoded thymidine kinase UL23 and then further converted to acyclovir triphosphate by cellular enzymes. This form is incorporated into the HSV viral DNA, halting further chain elongation due to the absence of a 3′-hydroxyl group, thereby preventing viral replication [32]. ACV is generally well tolerated, though rare adverse events can include nephrotoxicity (due to crystallization within renal tubules), neurotoxicity, disseminated intravascular coagulation, and pregnancy complications (owing to its ability to cross the placenta) [33]. Other FDA-approved antiviral medications, including penciclovir, famciclovir, ganciclovir, valacyclovir, cidofovir, and foscarnet, are currently used for the treatment of herpetic skin lesions. However, the limited efficacy of these antiviral medications is a significant concern, as they typically only shorten the recovery period by 1–2 days, with some patients experiencing no benefit [34]. Additionally, several natural remedies, such as lemon balm, propolis, and oral and topical zinc, have been shown to ameliorate HSV outbreaks and complications, with some demonstrating efficacy comparable to nucleoside analogs [35].

## 3. HSV Vaccines

Although efforts to develop an HSV vaccine date back to the 1920s [5], these initial attempts did not result in an effective candidate [36]. In 1999, a review article emphasized that an ideal prophylactic vaccine should induce sterilizing immunity, preventing all viral entry points into the human host, including the genital tract, nasal and oropharyngeal mucosa, and the eye, to avert both primary and latent infections [37].

Given that HSV-1 and HSV-2 share 80% genetic homology [38], their vaccine development should be approached together, focusing on therapeutic and prophylactic strategies. Prophylactic vaccines aim to provide immunity before exposure to the virus, while therapeutic vaccines target symptom reduction in already infected individuals. However, developing prophylactic vaccines is more complex due to ethical considerations and challenges in trial recruitment, resulting in a higher prevalence of therapeutic vaccine studies. Despite many candidates showing promise in early development stages, such as animal studies, these results have often lacked replication in human trials. For example, mice, guinea pigs, and even nonhuman primates such as rhesus macaques have demonstrated poor prediction of human prophylaxis, although *Cebus apella* monkeys were recently shown to be capable of mimicking human HSV-2 infection [39]. One major issue has been the lack of broad HSV antibodies and strong cell-mediated immune responses in vaccine recipients, despite high levels of specific neutralizing antibodies observed in clinical studies [40,41].

To address these challenges, the National Institutes of Health launched the 2023–2028 NIH Strategic Plan for HSV Research to achieve the longstanding goal of the therapeutic defeat of HSV [42]. Despite the difficulties, there are reasons for optimism. For example, varicella zoster virus (VZV), an alpha-herpesvirus closely related to HSV and responsible for shingles and chickenpox, is an effective live-attenuated vaccine available for prophylactic and therapeutic use [43]. The successful development of the human papillomavirus vaccine further demonstrates that an intramuscularly administered vaccine can effectively combat a mucosal genital viral pathogen [44]. The recent availability of full-length viral sequences for HSV-1 and HSV-2, coupled with an enhanced understanding of humoral- and cell-mediated immunity, holds promise for designing future candidate vaccines [45].

Various vaccine strategies are also under investigation, with subunit vaccines in different formats (cell culture-derived, recombinant, or modified live viruses) being the most studied. More promising approaches include live-attenuated, replication-competent, and replication-defective vaccines, as well as DNA vaccines, mRNA vaccines, and inactivated vaccines. These vaccines are at different development stages and are supported by biotechnology and pharmaceutical companies, academic institutions, and government agencies [46].

## 4. Vaccine Types

Table 1 shows all clinically relevant vaccines in articles screened from PubMed, EMBASE, Medline, and Google Scholar. These vaccines fall into the following five classes: subunit vaccines; live-attenuated replication-competent vaccines; live-attenuated replication-defective vaccines; DNA-based vaccines; and mRNA vaccines.

### 4.1. Subunit Vaccines

**GEN-003 (Genocea)**: GEN-003 (GEN-003/MM-2) is one of the best-known candidates for HSV therapeutic vaccines, containing the recombinant HSV antigens glycoprotein D (gD) and ICP4, along with a Matrix M-2 (MM) adjuvant. In preclinical studies, Skoberne et al. (2013) demonstrated GEN-003′s induction of broad-spectrum immune responses in mice and therapeutic efficacy in guinea pigs while decreasing recurrent shedding [47]. In 2015, GEN-003 phase 1 and 2 trials demonstrated significantly reduced genital lesions and viral shedding in over 310 participants [48]. Genital HSV-2 shedding was significantly reduced in all active vaccine groups, with a 60% reduction in the rate of genital lesions and elevated neutralizing antibody titers. Despite such promising results, Genocea ceased spending on GEN-003, shifting its focus to neoantigen cancer vaccines in 2017 [49,50].

**VCL-HB01 (Vical)**: Another therapeutic vaccine candidate that was recently abandoned was VCL-HB01, a DNA plasmid vaccine consisting of polynucleotides encoding codon-optimized gD2 and VP11/12 in combination with Vaxfectin, a lipid-based compound designed to enhance protein expression [51]. A phase 2 study conducted on 261 healthy HSV-2-seropositive adults with a self-reported history of at least 4 to 9 yearly recurrences did not meet its primary endpoint of reducing lesion recurrence rates despite the absence of serious adverse effects. Hence, VCL-HB01 represents another vaccine candidate targeting HSV glycoproteins that failed to progress after unsatisfactory phase 2 results [52].

**HerpV (Agenus)**: The HerpV therapeutic glycoprotein subunit vaccine is another HSV subunit vaccine that failed to progress past phase 2. HerpV (formerly called AG-707) consists of 32 HSV-2 peptides derived from 22 HSV-2 proteins that are non-covalently complexed to a heat shock protein 70 (HSP70) chaperone and formulated with a QS-21 saponin adjuvant [53]. Peptides for the vaccine, which include proteins spanning all classes of herpes proteins, were selected based on algorithms predicting human leukocyte antigen binding, synthesis feasibility, and proteasomal processing. Preliminary results from a phase 2 study of HerpV showed a 15% decrease in viral shedding, which persisted up to 6 months after the initial vaccine series [54].

**Simplirix (GlaxoSmithKline)**: This truncated glycoprotein D2 (gD2) vaccine candidate was tested in the phase 3 Herpevac Trial for Women [55]. Subjects were vaccinated with either the investigational vaccine (consisting of 20 μg of glycoprotein D2 from HSV-2 strain G in alum and 3-O-deacylated monophosphoryl lipid A as an adjuvant) or a control hepatitis A vaccine. Three doses of the vaccine were 58% protective against culture-positive HSV-1 genital disease but not protective against HSV-2 infection or disease [55]. Ultimately, it was concluded that the vaccine was unsuccessful in preventing HSV-2 infection or disease, as some women who became infected during the trial experienced recurrent disease [55].

### 4.2. Live-Attenuated Replication-Competent Vaccines

**HF10**: With decades of failure in subunit vaccines, recent focus has shifted to other vaccine candidates. Of these, the first to start development was HF10, a live-attenuated replication-competent HSV-1 naturally mutated vaccine (i.e., not engineered) for the genes *UL43*, *UL49.5*, *UL55*, and *UL56*, and latency-associated transcripts [56]. Immunization with HF10 protected mice against clinical symptoms elicited by HSV-2 inhibited HSV-2 replication at the site of virus introduction, reduced local inflammation, blocked neuroinvasion, and increased survival [57]. The protective effect of HF10 was also related to the induction of cellular immunity, mediated mainly by Th1 CD4+ T cells [57].

**0ΔNLS (RVx201) (from Rational Vaccines)**: 0ΔNLS (RVx201) is modified to be interferon sensitive and is thus unable to inhibit intracellular antiviral responses [58]. In preclinical studies, 0ΔNLS inhibited vaginal shedding in guinea pigs and could not establish latency in dorsal root ganglia, leading to the suppression of HSV reactivation [58]. The vaccine also conferred protective immunity to ocular HSV-1 challenge (pathology-free corneas) with reduced infection of the nervous system [59]. As a live vaccine, the 0ΔNLS vaccine is avirulent yet replication-competent and can elicit immunogenicity without producing clinical disease. This vaccine reactivates subclinically, eliciting immunogenicity against a wild-type virus, thus allowing the host to acquire protective immunity [60]. Nonclinical studies demonstrated that HSV-2 0ΔNLS is replication-competent, interferon-sensitive, and avirulent in immunocompetent mice, achieving an optimal balance between attenuation versus capacity to stimulate a protective immune response [61]. Such balance allows for better protection of its subjects from HSV compared with other vaccines, such as the leading candidate at that time (2011), the gD2 Herpevac trial vaccine [62]. Although one of the safety trials was controversial [63,64], in 2021, the vaccine received an Innovation Passport from the UK’s MHRA (Medicine and Healthcare Products Regulatory Agency)-led Innovative Licensing and Access Pathway (ILAP) [65]. Rational Vaccines, Inc. (Woburn, MA) also recently received a grant from the National Institutes for Allergy and Infectious Diseases for further vaccine studies [66].

**VC2 (Rational Vaccines)**: VC2 is a live-attenuated HSV-1 vaccine engineered to be incapable of entering neuronal axons. Such attempts to minimize side effects via engineering have been labeled rational [67]. VC2, which possesses deletions of gK aa31-68 and UL20 aa4-22, successfully protected against ocular immunopathogenesis in mice while preventing viral entry to neurons [68]. In addition, intramuscular administration to guinea pigs produced a transcriptional profile of Th17 and regulatory Tr1 responses [69].

### 4.3. Replication-Defective Vaccines

**HSV529 (Sanofi Pasteur)**: The HSV529 vaccine is a live HSV-1 virus depleted of UL5 and UL29 [70,71]. It was shown to elicit both humoral- and cell-mediated immunity, serum-neutralizing antibody titers, serum and vaginal antibodies to HSV-2 glycoprotein D, HSV2-specific antibody-dependent cellular cytotoxicity, and CD4+ and CD8+ T cell responses. Serum-neutralizing antibody titers significantly increased after three doses of HSV529, and this increase persisted for up to 6 months (*p* < 0.001), as did increased serum and vaginal antibodies to HSV-2 glycoprotein D (gD). In addition, the vaccine significantly induced HSV-2-specific antibody-dependent cellular cytotoxicity (*p* < 0.001). CD4+ T-cell responses were detected in 46%, 27%, and 36% of subjects in groups 1, 2, and 3, respectively. Concurrently, CD8+ T-cell responses were detected in 8%, 18%, and 14% of subjects in groups 1, 2, and 3, respectively [71].

89% of vaccine recipients experienced a mild-to-moderate injection site reaction versus 47% of placebo recipients. A total of 64% of vaccine recipients experienced systemic reactions versus 53% of placebo recipients. Two documented serious adverse events in two participants were concluded to be unrelated to HSV529 administration.

### 4.4. DNA Vaccines

**COR-1**: Cor-1 is a DNA vaccine consisting of two plasmids. One (codon optimized) codes for the HSV-2 envelope glycoprotein D (gD2), and the second has a truncated gD2 fused to ubiquitin [72]. In a preclinical model, Cor-1 induced a balanced adaptive humoral- and cell-mediated immune response in mice [73], while a phase 1 dose-escalating study showed safety and tolerability in 20 subjects.

**SL-V20 (SL VAXiGEN)**: SL-V20, a plasmid DNA vaccine against HSV2 glycoproteins gC, gD, and the UL39 ribonucleotide reductase, was 100% effective against mouse lethal challenge while also completely preventing vaginal infection. SL-V20 effects were T-cell-mediated, with B cells being dispensable to responses [74]. The current status of this vaccine is unclear.

### 4.5. mRNA Vaccines

**mRNA-1608 (Moderna)**: The COVID-19 pandemic ushered in the successful development of mRNA vaccines, which are now under study for multiple diseases, including Alzheimer’s disease and cancer. For HSV-2, Moderna, in collaboration with the University of Pennsylvania, has developed mRNA-1608-P101, a trivalent vaccine against HSV-2 gC2, gD2, and gE2 that is currently in a phase 1/2 trial for 365 patients aged 18 to 55 years (Clinicaltrials.gov ID NCT06033261). The trial’s duration is from September 2023 to June 2025. In preclinical studies, mRNA-1608 provided mice with 100% protection against lethal challenge [75] while also preventing dorsal root ganglion infection and inducing high titers of neutralizing antibodies and durable responses of CD4+ T follicular helper and memory B cells [76,77]. In rhesus macaques, the trivalent mRNA-1608 vaccine-induced neutralizing antibodies blocking gC2 and gE2 immune evasion, stimulated CD4 T cell responses, and elicited 100% protection in the vaginal challenge [78]. Comparison of the mRNA/nanoparticle formulation to baculovirus proteins with CpG/alum revealed day 2 and 4 vaginal cultures to be negative in 23 of 30 (73%) mice in the baculovirus group compared with 63 of 64 (98%) in the mRNA group [77]. In guinea pigs, 5 of 10 (50%) animals in the trivalent subunit protein group had vaginal shedding of HSV-2 DNA in 19 of 210 (9%) days, compared with 2 of 10 (20%) animals in the mRNA group that shed HSV-2 DNA in 5 of 210 (2%) days (*p* = 0.0052).

## 5. Conclusions and Future Directions

Humankind has contended with HSV viruses for millennia. As of 2024, significant strides have been made toward overcoming a virus that has persistently challenged our immune system throughout time. Our review identified 12 relevant vaccines, highlighting both progress and setbacks. Despite promising expectations, Genocea ceased the development of GEN-003 in 2017 to focus on neoantigen cancer vaccines. Similarly, HerpV, a subunit vaccine from Agenus, did not progress beyond favorable phase 1 results. 

On the other hand, there are vaccine candidates actively undergoing clinical trials, which include 0ΔNLS (a live-attenuated vaccine from Rational Vaccines), mRNA-1608 (a mRNA vaccine from Moderna), and COR-1 (a DNA-based vaccine from Anteris Technologies). Several vaccines have shown promising findings in preclinical studies as well. For example, the live vaccine 0ΔNLS demonstrated superior efficacy compared to gD2 vaccines (such as the Herpevac trial vaccine) in protecting guinea pigs from lethal challenges. Moderna’s mRNA vaccine has recently shown an 85% to 100% reduction in animal genital disease. Ongoing and future clinical trials will be necessary to evaluate the full efficacy and safety of HF10, VC2, and HSV 529 in humans.

Compared with subunit vaccines, other forms of vaccines, such as live-attenuated vaccines, hold promise for HSV due to the virus’ complex structures and ability to target multiple mechanisms of action, particularly when the virus is latent in neuronal cells. While these vaccines still need to be carefully evaluated in future clinical trials, the potential benefits they offer make them a promising avenue for further research and development.

## Figures and Tables

**Figure 1 viruses-16-01476-f001:**
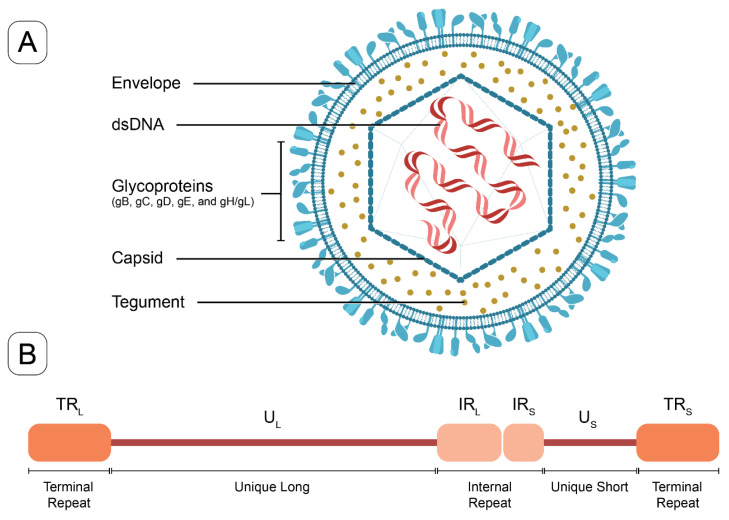
The structure of HSV virion and a schematic of HSV genome. (**A**) The HSV virion consists of a linear double-stranded DNA genome enclosed within an icosahedral capsid. Tegument proteins are distributed between the capsid and the lipid bilayer envelope. Glycoproteins are embedded in the outer surface of the envelope, facilitating viral entry (key glycoproteins in vaccine targeting include gB, gC, gD, and gE). (**B**) The HSV-1 genome schematic includes unique long (U_L_) and unique short (U_S_) regions, which are flanked by internal or terminal repeats, (IR_L_ and TR_L_) and (IR_S_ and TR_S_).

**Figure 2 viruses-16-01476-f002:**
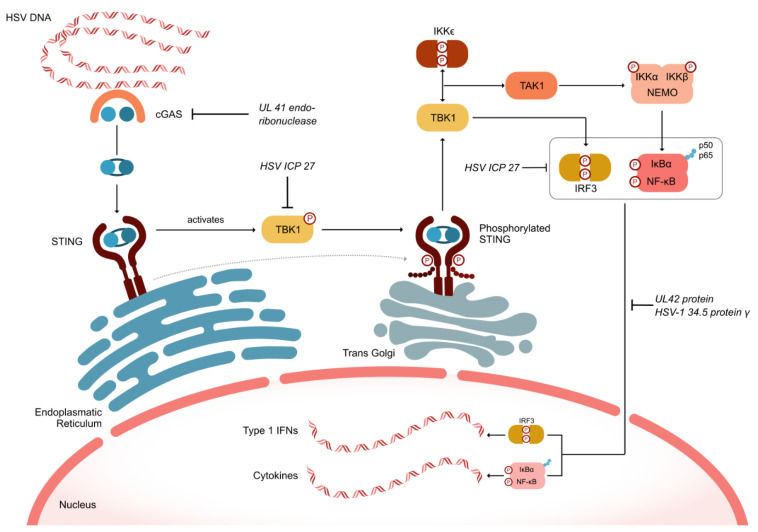
HSV defense mechanisms to evade intrinsic and adaptive immune responses. cGAS (cyclic GMP-AMP synthase); IFN (interferon); IKK (inhibitor of κB kinase); IRF3 (interferon regulatory factor 3); NEMO (nuclear factor-kappa B essential modulator); NF-κB (Nuclear Factor-kappa B); IκBα (NF-κB inhibitor alpha); STING (stimulator of interferon genes); TAK1 (transforming growth factor-β-activated kinase 1; TBK1 (tank-binding kinase 1).

**Table 1 viruses-16-01476-t001:** Clinically relevant HSV vaccines in articles screened from PubMed, EMBASE, Medline, and Google Scholar.

Vaccine Type	Vaccine Candidate	Antigen	Adjuvant/Route of Administration	Attenuation	Status	Results	Refs
Subunit	GEN-003 (from Genocea)	gD2/ICP4	Matrix M2; arm muscle injection		Inactive	Unsatisfactory phase 2 results. Further development was abandoned.	[47,48,49,50]
	VCL-HB01 (Vical)	gD2 and VP11/12	injection		Inactive	Unsatisfactory phase 2 results. Further development was abandoned.	[51,52]
	HerpV (from Agenus)	32 peptides + HSP70 chaperone	QS-21; injection		Inactive	Reported to reduce viral shedding in HSV-2-infected subjects by approximately 15% after boost immunization.	[53,54]
	Simplirix (GlaxoSmithKline; Herpevac Trial for Women)	HSV-2 gD2	MLA; injection		Inactive	Three vaccine doses were 58% protective against culture-positive HSV-1 genital disease but did not protect against HSV-2.	[55]
Live-attenuated replication-competent	HF10	A live-attenuated, replication-competent HSV-1 naturally mutated (i.e., not engineered)	Injection	Naturally mutated for *UL43*, *UL49.5*, *UL55*, *UL56*, *LAT* genes	Preclinical	Immunization with HF10 protected mice against clinical symptoms elicited by HSV-2 inhibited HSV-2 replication at the site of virus introduction, reduced local inflammation, blocked neuroinvasion, and increased survival.	[56,57]
	0ΔNLS (RVx201) (Rational Vaccines)	ICP0	Alum, lipid A; injection	Targeted mutations within full-length ICP0	Phase 1/2	Superior effectiveness of gD2 vaccines at protecting guinea pigs from lethal challenges. Conferred immunity to ocular HSV-1 challenge (pathology-free corneas) with reduced infection of the nervous system. Preclinical trials show good efficacy.	[58,59,60,61,62,63,64,65,66]
	VC2 (HSV-1) (Rational Vaccines)		Injection	Deletions of gK aa31-68 and UL20 aa4-22	Preclinical	Decreased clinical severity of acute and recurrent HSV-2 disease and shedding. Decreased dorsal root ganglion viral load in guinea pigs.	[67,68,69]
Replication-defective	HSV529 (Sanofi Pasteur)		None; intramuscular injection	Deletions of UL5 and UL29	Phase 2	Phase 1: 64% of vaccine vs. 53% of placebo recipients had systemic reactions. Serum and vaginal antibodies to HSV-2 glycoprotein D (gD) significantly increased after 3 doses.	[70,71]
DNA	COR-1 (Anteris Technologies, formerly Admedus)	Codon-modified full-length HSV-2 gD2 and ubiquitin-fused truncated gD2	Vaxfectin; injection		Phase 2	Achieved safety, tolerability, and immunogenicity phase I endpoints. While antibody responses were not observed, cell-mediated immunity was demonstrated.	[72,73]
	SL-V20 (SL VAXiGEN)	HSV-2 gB, gD2, and UL39 plasmid DNA	Cytokines; injection		Preclinical	100% mouse survival from lethal challenge. T cell-mediated protective response (B cells were dispensable).	[74]
mRNA	mRNA-1608-P101 (Moderna)	HSV-2 gC2, gD2, gE2	Lipid nanoparticles; injection		Phase 1/2	100% mouse and guinea pig survival upon lethal challenge. Reduced animal genital disease by 85–100%. Outperformed a protein form of the same vaccine.	[75,76,77,78,79]

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
