# Peer review of "Toward the Eradication of Herpes Simplex Virus: Vaccination and Beyond"

_viruses, 2024, doi:10.3390/v16091476_

Round 1
Reviewer 1 Report
Comments and Suggestions for Authors
In the review manuscript, the authors have compiled an impressive overview of the literature related to the design, development and efficacy of vaccines against Herpes Simplex Viruses types 1 and 1 (HSV-1 and HSV-2, respectively). The authors have done an outstanding job of describing the different types of formats utilized as platforms for anti-HSV vaccines, and describe in detail the outcomes of pre-clinical and clinical trials to date. The authors also provide a strong argument for the future direction of the research to develop more effective vaccines in the future.
One major issue with the manuscript in its present form is the description of the HSV genome, where the authors state that the genome consists of a Unique Long and Unique Short segment, containing different numbers of haploid genes. However, they neglect to include the Internal Repeat and Terminal Repeat regions, which contain diploid genes, many of which the authors mention in the design of the different replication competent and replication incompetent forms used for vaccines. It is suggested that the authors expand this section and also include a diagram of the linear form of the HSV genome (HSV-1 would be sufficient) as part of a revised Figure 1.
A minor issue is regarding the mRNA vaccines. While the authors explain very well the preclinical findings of the studies from the University of Pennsylvania/Moderna group, and the ongoing clinical studies, they omit the fact that some preclinical studies were also performed in non-human primates, which is an important pre-clinical study. The authors should include this element in the review.
Author Response
In the review manuscript, the authors have compiled an impressive overview of the literature related to the design, development and efficacy of vaccines against Herpes Simplex Viruses types 1 and 1 (HSV-1 and HSV-2, respectively). The authors have done an outstanding job of describing the different types of formats utilized as platforms for anti-HSV vaccines, and describe in detail the outcomes of pre-clinical and clinical trials to date. The authors also provide a strong argument for the future direction of the research to develop more effective vaccines in the future.
We greatly thank the reviewer for the kind comments that we have done an outstanding job of describing different vaccine formats and provided a strong argument for future research directions.
One major issue with the manuscript in its present form is the description of the HSV genome, where the authors state that the genome consists of a Unique Long and Unique Short segment, containing different numbers of haploid genes. However, they neglect to include the Internal Repeat and Terminal Repeat regions, which contain diploid genes, many of which the authors mention in the design of the different replication competent and replication incompetent forms used for vaccines. It is suggested that the authors expand this section and also include a diagram of the linear form of the HSV genome (HSV-1 would be sufficient) as part of a revised Figure 1.
We thank the reviewer for the helpful comment. We have now revised Figure 1 to depict a linear HSV genome, including the IRS (internal repeat) and TRS (terminal repeat) regions, in addition to the UL and US segments, containing diploid genes that encode vaccine targets. We also now expand Section 3, "HSV life cycle, pathogenesis, and immune evasion," to discuss the IRS and TRS regions of the HSV genome (page 6, lines 5 - 6).
A minor issue is regarding the mRNA vaccines. While the authors explain very well the preclinical findings of the studies from the University of Pennsylvania/Moderna group, and the ongoing clinical studies, they omit the fact that some preclinical studies were also performed in non-human primates, which is an important pre-clinical study. The authors should include this element in the review.
We now discuss primate preclinical vaccine studies in Section 5.5, "mRNA vaccines, mRNA-1608 (Moderna)", citing Awashti et al., 2017, PLoS Pathogens 13(1):e1006141 (page 23, lines 4 - 6). We also discuss the use of C. apella monkeys as a model system for predicting human efficacy of prophylactic vaccines (Section 4, "HSV Vaccines", page 13, lines 11 - 14), citing Wang et al., 2024, PLoS Pathogens 20(9):e1012477.
Reviewer 2 Report
Comments and Suggestions for Authors
The manuscript by Chang JY et al. offers a thorough and detailed review of HSV vaccines.
It serves as an invaluable resource for many researchers in the field, presenting an impartial summary of various aspects of HSV biology, related pathologies, and the complexities of HSV vaccination. Therefore, I believe this paper holds significant value for publication and requires only minor revisions, as outlined below.
span style="mso-list: Ignore"-span style="font: 7.0pt 'Times New Roman'" /span/span/spanKeywords: Consider revising to include terms such as herpes simplex virus type 1, live attenuated vaccines, mRNA vaccine, replication-defective strains, interferon response.
span style="mso-list: Ignore"-span style="font: 7.0pt 'Times New Roman'" /span/span/spanLine 69: Place the comma after the quotation marks.
span style="mso-list: Ignore"-span style="font: 7.0pt 'Times New Roman'" /span/span/spanFigure 1 legend: Revise the legend to "Herpes simplex virus virion." The figure is somewhat oversimplified; consider adding more details, such as key envelope proteins (e.g., gC, gD, gE).
span style="mso-list: Ignore"-span style="font: 7.0pt 'Times New Roman'" /span/span/spanFigure 2: Check the figure resolution, as it appears blurred.
span style="mso-list: Ignore"-span style="font: 7.0pt 'Times New Roman'" /span/span/spanTable 1: Include the route of administration.
Author Response
The manuscript by Chang JY et al. offers a thorough and detailed review of HSV vaccines.
We thank the reviewer for remarking that we offer a thorough and detailed review of HSV vaccines.
Keywords: Consider revising to include terms such as herpes simplex virus type 1, live attenuated vaccines, mRNA vaccine, replication-defective strains, interferon response.
We now include these terms in the manuscript keywords (page 3, lines 4 - 5).
Line 69: Place the comma after the quotation marks.
We have now placed all instances of punctuation after quotation marks.
Figure 1 legend: Revise the legend to "Herpes simplex virus virion." The figure is somewhat oversimplified; consider adding more details, such as key envelope proteins (e.g., gC, gD, gE).
Thank you. We have now added more details to Figure 1 and its legend, including the key glycoproteins in vaccine targeting (page 7, lines 5 - 6).
Figure 2: Check the figure resolution, as it appears blurred.
We believe this the result of pasting the figure into Word and scaling it. We have now re-inserted the figure in the manuscript. Also, please see Figure 2 as an attachment separate from the manuscript.
Table 1: Include the route of administration.
The route of administration is now included in Table 1.